# Dielectric anomalies and interactions in the three-dimensional quadratic band touching Luttinger semimetal $Pr_2Ir_2O_7$

Bing Cheng[1], T. Ohtsuki[2], Dipanjan Chaudhuri[1], S. Nakatsuji[2,3], Mikk Lippmaa[2] & N.P. Armitage[1]

Dirac and Weyl semimetals with linearly crossing bands are the focus of much recent interest in condensed matter physics. Although they host fascinating phenomena, their physics can be understood in terms of weakly interacting electrons. In contrast, more than 40 years ago, Abrikosov pointed out that quadratic band touchings are generically strongly interacting. We have performed terahertz spectroscopy on the films of the conducting pyrochlore $Pr_2Ir_2O_7$, which has been shown to host a quadratic band touching. A dielectric constant as large as $\tilde{\varepsilon}/\epsilon_0 \sim 180$ is observed at low temperatures. In such systems, the dielectric constant is a measure of the relative scale of interactions, which are therefore in our material almost two orders of magnitude larger than the kinetic energy. Despite this, the scattering rate exhibits a $T^2$ dependence, which shows that for finite doping a Fermi liquid state survives—however, with a scattering rate close to the maximal value allowed.

[1] The Institute for Quantum Matter and the Department of Physics and Astronomy, The Johns Hopkins University, Baltimore, MD 21218, USA. [2] Institute for Solid State Physics The University of Tokyo, Kashiwa 277-8581, Japan. [3] CREST Japan Science and Technology Agency, Kawaguchi, Saitama 332-0012, Japan. Correspondence and requests for materials should be addressed to N.P.A. (email: npa@pha.jhu.edu)

Zero-gap semimetals are an extensively investigated area of modern condensed matter physics. With the advent of graphene[1] and topological insulators[2,3], linear band crossings in two dimensions (2D) have been shown to be a source of much interest in physics. Moreover, three-dimensional (3D) materials with linear band crossings in the form of topological (Weyl) and related (massless Dirac) materials exist and are a very active subject of current investigation[4]. Although the physics here is fascinating, these are generally weakly interacting systems that can be understood within the context of free fermion theories[5]. However, other zero-gap semimetal possibilities exist. One is bilayer graphene which has a 2D quadratic band touching (QBT)[6] and is predicted to host a variety of interesting interacting phases[7]. $\alpha$-Sn and HgTe are well-known older materials that possess a 3D QBT (Fig. 1) at the zone center in their fully symmetric cubic state. The crossing is protected—as it is in the 3D massless Dirac case—by point group and time reversal ($\mathcal{T}$) symmetries; their valence and conduction bands belong to the same irreducible representation of the symmetry groups. These systems can be described in a minimal band structure by the Luttinger Hamiltonian for inverted gap semiconductors[8]. The 4-fold degeneracy at the touching point cannot be removed unless the symmetries are broken. This has been of renewed interest due to the fact that under uniaxial strain or in a superlattice geometry, such systems can become gapped topological insulators[9].

A number of interesting effects are expected in these 3D QBT Luttinger semimetal (LSM) systems. Electronic correlations are predicted to be more pronounced than that in the linear band crossing systems due to the rapidly increasing density of states. When the Fermi energy ($E_F$) is tuned to zero, one expects a divergent complex dielectric constant because of the vanishing threshold for interband transitions[10,11]. Random phase approximation (RPA) calculations that include interactions at the lowest order give a contribution where both components go as $\sim 1/\sqrt{\omega}$[12]. This divergence is expected to be cutoff by the finite $E_F$, which exists due to impurity doping in all real materials. As shown in the seminal work by Abrikosov and Beneslavskii (AB) in the 1970s[13,14], in the vicinity of the band touching, LSMs are expected

to be strongly interacting and the concept of quasiparticles is inapplicable at energies well below the scale of the dominating electron–hole interaction. This energy scale is set by the excitonic correlation energy $\left( E_0 = \frac{\mu e^4}{32\pi^2 \varepsilon_\infty^2 \hbar^2} = 13.6\,\text{eV}\, \frac{\mu/m_0}{(\varepsilon_\infty/\epsilon_0)^2} \right)$. Here $\mu$ is the reduced mass of the conduction–valence band system, $m_0$ is the free electron mass, $\varepsilon_\infty$ is the background dielectric constant due to all excitations not associated with the quadratic touching bands (e.g., phonons and higher energy bands), and $\epsilon_0$ is the vacuum permittivity. As discussed below, $E_0 \sim 0.41$ eV in the current system. Taking advantage of the inherent scale-free criticality in such a system and using an $\epsilon$-expansion about 4 spatial dimensions, Abrikosov derived scaling exponents for various observables[15]. AB's work was a remarkable demonstration almost 50 years ago of the possibility of a non-Fermi liquid. More recently, Moon et al.[16] showed that the long-range electron–electron interactions may generically stabilize a non-Fermi liquid phase, rather than driving the system to an instability. This stability may be understood as a balance of the screening of Coulomb interactions by electron–hole pairs and mass enhancement of the quasiparticles dressed by the same virtual pairs. In contrast, it has been argued recently that in 3D and for the single band touching found in known materials, the LSM phase is unstable at the low energies due to forming a gapped nematic[17,18] or $\mathcal{T}$ breaking phase[19]. In principle, all such interaction-driven phenomena could exist in the classic LSMs HgTe and $\alpha$-Sn. However, such effects have never been observed, presumably because the broad bands in such compounds decrease the relative scale of the electronic correlations ($E_0$) and the finite chemical potential $E_F$ given by residual doping has been sufficient to cutoff the divergence of the dielectric constant that is associated with the zero-energy interband transitions.

Recently, it has been shown in a photoemission study[20] that the pyrochlore oxide $Pr_2Ir_2O_7$ (Pr227) possesses a 3D QBT at its Brillouin zone center. Pr227 is a very interesting material with a rich phenomenology[21,22]. Different from other pyrochlore iridates, Pr227 is a metal that does not show any sign of magnetic dipole order down to 100 mK, but does show a large anomalous Hall effect below 50 K. Moreover, a non-zero Hall conductivity is also observed even without an external magnetic field applied, which has been proposed to be related to a long-range scalar spin chiral order[23]. A QBT in Pr227 is believed to be formed between $J = 3/2$ bands in essentially the same fashion as the classic systems[16]. Importantly, however, the effective band masses were found with a photoemission of approximately $6.3m_0$[20], which is almost 300 times larger than in $\alpha$-Sn[24]. This enhances the relative role of interaction, making $E_0 \sim 0.41$ eV in this material and opens the possibility of probing the strongly interacting regime.

In this work, we characterize the low-frequency electrodynamic response of Pr227 thin films by THz spectroscopy and find it consistent with the phenomenology expected for a slightly doped LSM. Although broad at high temperatures, in both samples investigated, a relatively narrow Drude peak appears below 80 K. By using the Drude model to fit the real and imaginary parts of the conductivity, we extract the temperature-dependent plasma frequency, scattering rates, and low-frequency dielectric constant $\tilde{\varepsilon}$. We observe that $\tilde{\varepsilon}/\epsilon_0$ has a strong temperature dependence and becomes as large as 180 at low temperature. The temperature-dependent scattering rate of the Drude contribution below 80 K shows a $T^2$ law and may be indicative of a Fermi liquid, but with a coefficient that is close to the maximum allowed value. Taken together, these different aspects can be self-consistently modeled as an LSM with a finite $E_F$.

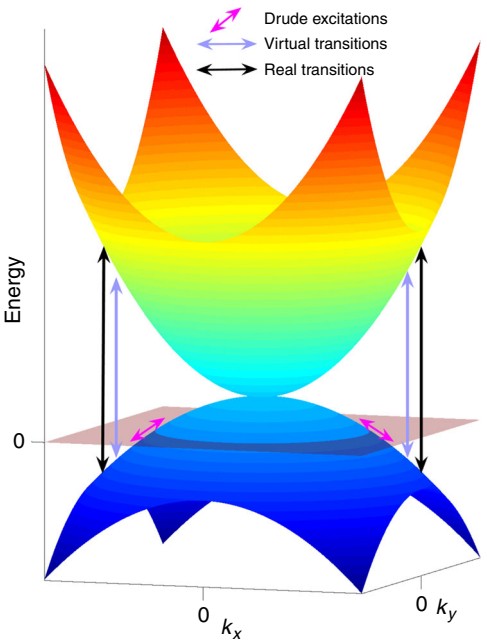

**Fig. 1** Schematic of quadratic band touchings in a system that is slightly doped to give a finite $E_F$. One can distinguish the contribution of low-energy Drude excitations near $E_F$ as well as virtual and real interband transitions

## Results

**dc transport.** In Fig. 2, we show the dc resistivity of two different samples S1 and S2 as a function of temperature taken in a four-

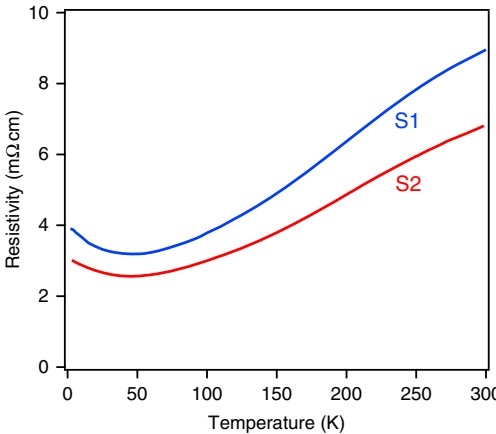

**Fig. 2** dc resistivity of samples S1 and S2. Geometric factors of these data were calibrated assuming that the optical conductivity measured by TDTS at 150 K was independent of frequency up to 1 THz

probe geometry on a square sample. Geometric factors for the resistivity were calibrated assuming that the optical conductivity measured by TDTS at 150 K was independent of frequency up to 1 THz. One can see that with decreasing temperature, the resistivities of both Pr227 films show metallic behavior down to 60 K. Below 60 K, the resistivity shows an upturn and increases with further cooling. These data are very similar to the resistivity obtained from single crystals, except that the minimum is even more enhanced[21]. This low-temperature upturn of resistivity has been previously interpreted to arise from Kondo scattering between the conduction electrons of Ir atoms and localized magnetic moments of Pr atoms[22]. However, in this work, we show that the minimum in the resistivity is a consequence of the interplay between a decreasing scattering rate and a decreasing charge density when cooling in a slightly doped 3D LSM.

**THz spectroscopy**. In Fig. 3a, b, we show the THz range complex conductivity of sample S1 as a function of frequency at a number of temperatures. Sample S2 showed a similar phenomenology. Consistent with the resistivity measurement, the overall scale of the THz conductivity first increases with decreasing temperature, and then decreases with further cooling below 60 K. At temperatures above 150 K, the spectra are relatively flat, which means that the scattering rates of the carriers are large as compared to the considered spectral region. Below 60 K, although the real part of optical conductivity spectra decreases with cooling, the trend of a negative slope of the spectra increases, which indicates that the scattering rates of the carriers are decreasing. The real part of the optical conductivity spectra can be easily fit by a Drude expression $\frac{\epsilon_0 \omega_p^2 \tau}{1 - i\omega\tau}$ with a temperature-dependent scattering rate $(1/2\pi\tau)$ and plasma frequency $\omega_p^2 = \frac{ne^2}{\epsilon_0 m^*}$. We use the conductivity from the dc measurements (symbols at $\omega = 0$) to constrain the Drude fit. The scattering rates and the plasma frequency from the fits for S1 and S2 are shown in Fig. 4a, b. The plasma frequency shows an approximately linear function of temperature.

The scattering rate monotonically increases with increasing temperature. Note that since the real part of the conductivity is flat at temperatures much above 90 K, we cannot fit the data to obtain $\omega_p$ and $1/\tau$ separately above this temperature. To continue the fits, in this region, we assume that the linear dependence of the plasma frequency continues for another factor of 1.5 in temperature. Although one can see that with this assumption the functional dependence of $1/\tau$ continues, we only use data below 80 K for further analysis below. One can see that in the temperature region of the resistivity minimum, the plasma frequency and scattering rates do not show any anomaly. This is a strong evidence against Kondo scattering as the source of the minimum. The low temperature value of the plasma frequency allows us to determine $E_F$. Using the above relation for the plasma frequency and the effective mass of the conduction band $m^* = 6.3$, $m_0$ determined by photoemission[20], at the lowest temperatures, we find $E_F$'s of $7 \pm 1$ meV and $12 \pm 1$ meV for S1 and S2, respectively. These values are close to the $E_F$ of 17 meV determined by the analysis of the anomalous Hall effect on a third film that had a 50% higher residual resistivity (all analysis details are given in the Supplementary Note 1).

Interestingly, the imaginary parts of optical conductivity cannot be fit by using only a Drude term with the same parameters. In addition to the Drude term, one must subtract a large imaginary contribution $\omega\tilde{\varepsilon}$, where $\tilde{\varepsilon}$ is a background dielectric constant that arises from virtual excitations at energies above the measured spectral range (Fig. 1). To see this more clearly, we show two comparative fits in Fig. 3c, d. At 150 K, a broad Drude term with a small $\tilde{\varepsilon}$ can fit the real and imaginary parts simultaneously. However, at 6 K, unless we add a very large $\tilde{\varepsilon}$ term, one cannot fit real and imaginary parts simultaneously with the same parameters. In Fig. 3c, we also show a fit with a more conventional $\tilde{\varepsilon}$ (blue dashed line). It deviates from the measured imaginary part of conductivity considerably. The temperature dependence of $\tilde{\varepsilon}/\epsilon_0$ is given in Fig. 4c for both samples S1 and S2. At low temperature, we find that $\tilde{\varepsilon}/\epsilon_0$ can be as large as $180 \pm 10$ for S1 and $120 \pm 10$ for S2, which as discussed below should be considered as very large values.

**Discussion**

The above results can be interpreted self-consistently if Pr227 is a 3D QBT system. Earlier calculations have predicted that Pr227 is a zero-gap semiconductor with quadratic band dispersion[16], which has been subsequently demonstrated by photoemission[20]. Unlike conventional metals, the charge density in such a system can be strongly temperature dependent over a large temperature range because the threshold to excite thermal carriers is low even if the Fermi level is not exactly at, but just close to, the touching point of conduction and valence bands. The competition of temperature dependences between plasma frequency and scattering rates can easily result in a minimum of the dc resistivity at finite temperature.

The complex conductivity is related to the complex dielectric function as $\varepsilon(\omega) = 1 + i\sigma(\omega)/\omega$. At the frequencies of interest, one may in principle expect at least three distinct contributions to the dielectric function of an LSM e.g., $\varepsilon = \varepsilon_{Drude} + \varepsilon_{QBT} + \varepsilon_\infty$. Here $\varepsilon_{Drude}$ is a small metallic contribution that is finite for non-zero temperature or doping, $\varepsilon_{QBT}$ is the contribution from the QBT, and $\varepsilon_\infty$ is again the contribution to the dielectric constant from all transitions not ascribed to the QBT bands. In our fits, $\varepsilon_{QBT} + \varepsilon_\infty$ is accounted for by $\tilde{\varepsilon}$. For $E_F = 0$, the QBT gives a divergent contribution to the dielectric function whose real and imaginary parts arise from virtual and real excitations, respectively, between bands. Within the simplest RPA theory[10–12] and in the limit of zero temperature and with $E_F = 0$, the QBT contribution to the dielectric function is

$$\varepsilon_{QBT}(\omega) = \epsilon_0 \sqrt{\frac{2\mu e^4}{\varepsilon_\infty^2 \hbar^3 \omega}} [1 + i]. \qquad (1)$$

Note that Eq. (1) can be written as $8\pi$ times the square root of the ratio of the effective excitonic energy scale ($E_0$) to the excitation energy ($\hbar\omega$). It is also interesting to note the identical form of $\varepsilon'_{QBT}$ and $\varepsilon''_{QBT}$. A finite $E_F$ cuts off the divergences associated with Eq. (1). In that case, the imaginary part of $\varepsilon_{QBT}$ is multiplied

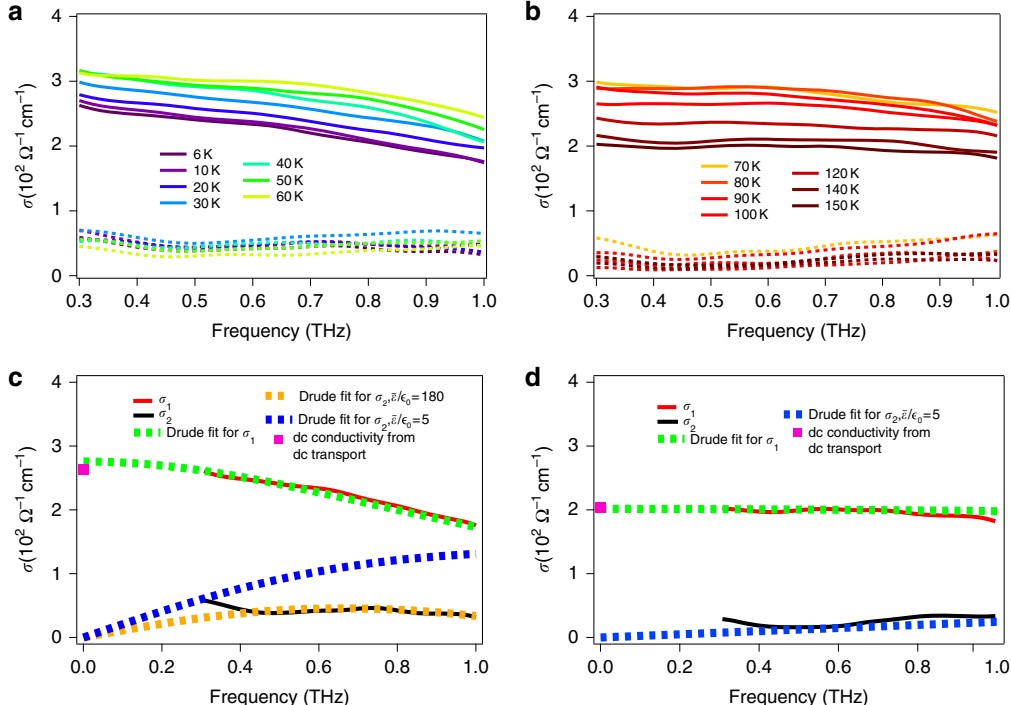

**Fig. 3** THz range optical conductivity. **a**, **b** THz range optical conductivity for real (solid line) and imaginary parts (dashed line) of the conductivity for sample S1 in two different temperature ranges. **c**, **d** Fits of optical conductivity at 6 and 150 K with constraints of dc conductivity from dc transport

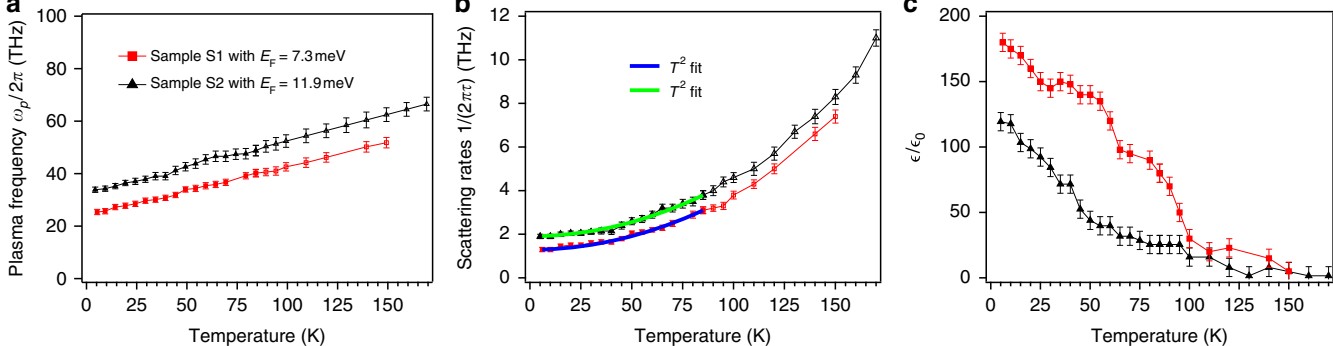

**Fig. 4** Drude model fit parameters. **a** The temperature-dependent plasma frequency from the Drude fit. Closed and open markers represent, respectively, the results of fits where the plasma frequency was unconstrained or constrained to a linear dependence as described in the text. **b** The temperature-dependent transport scattering rate from Drude fit. Scattering rates ($1/2\pi\tau$) below 80 K are fit to the functional form $\frac{1}{2\pi\tau_0} + AT^n$ and $n$ is extracted to be $2 \pm 0.2$ for both samples. **c** The temperature-dependent dielectric constant $\tilde{\varepsilon}/\epsilon_0$ from the Drude fit. Error bars are estimated as parameter range where acceptable fits (<4% difference from the data over the spectral range) to $\sigma$ are obtained

by the step function $\Theta(\hbar\omega - \gamma E_F)$, where $\gamma = \left(1 + \frac{m_v}{m_c}\right)$. Due to Pauli blocking, finite $E_F$ cuts off the divergence of virtual excitations at an energy $\gamma E_F$ that controls the real part of $\varepsilon_{QBT}$ at low $\omega$. One can find $\varepsilon'_{QBT}$ through a Kramers–Kronig transformation. Due to the sharp cutoff in $\varepsilon''$, $\varepsilon'$ is logarithmically divergent at $\gamma E_F$; however for $\hbar\omega \ll \gamma E_F$, $\varepsilon'_{QBT}$ can be found by letting $\hbar\omega \to \frac{\pi^2}{16}\gamma E_F$ giving $\varepsilon'_{QBT}(\hbar\omega \ll \gamma E_F) = 32\epsilon_0\sqrt{\frac{E_0}{\gamma E_F}}$. (The full expression and motivation for the prefactor is in the Supplementary Note 2.)

Using Eq. (1) with $\mu = 0.5 \, m^*$, the $E_F$'s determined from the Drude spectral weight, and the $\varepsilon_\infty/\epsilon_0 \approx 10$ found in other pyrochlores[25], we predict $\varepsilon'_{QBT}/\epsilon_0$ at low $\omega$ to be around 170 for S1 and around 132 for S2. The excellent agreement with observed values perhaps should be considered fortuitous, considering the uncertainty in the dielectric constant $\varepsilon_\infty$ and reduced mass.

However, note that in classic LSMs such as $\alpha$-Sn and HgTe, the contribution from the QBT has been determined to be far smaller ($\tilde{\varepsilon}/\epsilon_0 \sim 3.5$ and 7, respectively)[24,26]. In conventional semiconductors (Si or GaAs) and oxide insulators, the total dielectric constant in the THz range is typically found to be of order 10. Note that with the $\hbar\omega \to \frac{\pi^2}{16}\gamma E_F$ substitution, and $\mu^* = 0.5m^*$, one can express Eq. (1) as 17.5 times the square root of the characteristic scale for electron–hole interactions ($E_0$) to the average kinetic energy $\frac{3}{5}E_F$. With the values found at low temperature, for S1 one finds a large relative scale for interactions almost 100 times larger than the kinetic energy.

It is natural to ascribe the large measured value of $\varepsilon$ to the near divergence of the dielectric constant in a 3D QBT; however, one must be careful in the quantitive application of Eq. (1). When further including interactions in a high-order calculation of the dielectric function, the result from RPA Eq. (1) actually appears

as only the first term in an expansion in the parameter $E_0/E_F$. If this parameter is large—as it is in our case—RPA may not formally hold. Indeed, the breakdown of perturbation theory is what compelled AB to develop their scaling theory[13,14] in which strong interactions are believed to drive the form of the dielectric constant into a regime where the dynamic exponent $z$ differs slightly from 2. Interestingly, the RPA still provides a surprisingly robust starting point for understanding our results. The AB expression is $\varepsilon = \epsilon_0 \left(\frac{\omega_0}{\omega}\right)^{1-1/z}$, where $\omega_0$ is defined such that this expression reduces to Eq. (1) if $z=2$. To find the scale of the further terms in the $\sqrt{E_0/E_F}$ expansion, we work backwards from AB's expression to first order in $1/z$ with an expansion around $z=2$ to get $\epsilon_0 \left(\frac{\omega_0}{\omega}\right)^{1-1/z} \sim \sqrt{\frac{\omega_0}{\omega}} \left(1 - \delta \ln \frac{\omega_0}{\omega}\right)$, where $\delta = \left(\frac{1}{z} - \frac{1}{2}\right)$. From previous calculations, $\delta$ can be estimated[16] to be approximately 0.055 and therefore (upon substituting for $E_F$) corrections to the RPA form are estimated to only be of order 0.3%. Although the system is in a strongly interacting regime, the consequences of strong interactions are subtle.

With increasing temperature, charges are excited and as shown in Fig. 4a, the plasma frequency increases. These thermal carriers block low-energy interband transitions around the node, which weakens the enhancement of the dielectric constant observed at lower temperature. When the temperature is raised beyond the degeneracy temperature, RPA calculations predict that the dielectric constant is expected to fall off as $\sim 1/\sqrt{T}$[10]. However, one can see in Fig. 4c that dielectric constant decays much faster. One explanation of this discrepancy between experiment and theory may come from the fact that the quasiparticle spectral function of Pr227 as measured by photoemission has an extremely strong temperature dependence. Near the band touching, quasiparticle-like features were only observed below 100 K[20]. If the sharp LSM quasiparticle spectrum of the system gradually loses its features as temperature increases, it is reasonable to observe faster decay of the dielectric constant.

The temperature dependence of the scattering rate is also interesting. As seen in Fig. 4b, the scattering rates for samples S1 and S2 have very similar temperature dependences with only different offsets that are likely to come from impurity scattering. It is interesting to note the effects of impurities can be accounted for self-consistently when comparing S1 and S2. Compared to S2, S1 has a lower $E_F$ (as determined from its spectral weight of the Drude peak), higher $\varepsilon_{QBT}$, and lower residual scattering. Regarding the temperature dependence, we fit the scattering rates of the spectra with the functional form $1/2\pi\tau_0 + AT^n$ up to the temperature scale of 80 K, where quasiparticle-like peaks disappear in the ARPES spectra. Note that this expression assumes that a Matthiessen's-like rule applies to the scattering rate despite the appreciable disorder levels. The parameter $n$ is found to be 2 ± 0.2 for both samples. When holding $n$ at 2, the coefficient $A$ of the $T^2$ term of both fits are approximately of the same value [7 (meV)$^{-1}$]. This suggests that the temperature dependence of scattering rates is not sensitive to impurity density and charge doping even though $E_F$ changes between samples by about 35%. There are a few possibilities to explain this temperature dependence. The first, which was considered in the classic zero-gap semiconductor systems[10,11], is that the elastic scattering itself is temperature dependent. This is, of course, quite unlike the situation in normal metals. But in an LSM, the dielectric constant is strongly $T$ dependent, which leads to $T$-dependent screening that in principle can lead to a temperature-dependent elastic scattering. However, in this scenario, one would expect that the coefficient of the temperature dependent term would scale with the $T=0$ residual term. This is not observed.

The other possibility is that the $T^2$ behavior is indicative of Fermi-liquid-like physics. As discussed above, in a 3D LSM such as Pr227, a non-Fermi liquid phase is expected to be stabilized at

zero $E_F$ via the balance of the screening of Coulomb interactions by electron–hole pairs and the mass enhancement of the quasiparticles dressed by these pairs[13,14,16]. However for finite $E_F$, a Fermi liquid can be stabilized. Our data gives evidence for the fact that, when being probed at a frequency $\omega$ for $\hbar\omega \ll k_B T \ll E_F \ll E_0$, although interactions remain incredibly strong, their character is not such as to destabilize the Fermi liquid. In general one expects that the inelastic scattering for a Fermi liquid is bounded by the limit $\frac{1}{\tau_{in}} \sim \frac{T^2}{E_F}$. We find find that the coefficients of $T^2$ are very close to the value given by the independently measured values of $E_F$, showing that due to the strong interactions the scattering here essentially saturates this bound.

There are very interesting recent proposals for 3D LSMs in that, besides the long-range Coulomb interaction, the short-range interaction may also play an important role in the electronic structure[17–19]. Especially in the case of a 3D LSM that has just a single touching point, this short-range Coulomb interaction is predicted to destroy the non-Fermi liquid state stabilized by long-range Coulomb interaction and phases such as a Mott insulating state can appear. According to theory, the Mott gap in Pr227 may be estimated to be of order 4 meV. However, we see no sign of any such gapping or incipient gapping. It may be that finite $E_F$ removes this instability or significant anisotropy in the band structure and restores stability of the LSM as predicted[27], or perhaps the role of rare-earth Pr spins needs to be accounted for.

We have studied the THz-range optical responses of Pr227 thin films. We find that its low-energy dielectric constant is anomalously large which is reasonably ascribed to the virtual fluctuations in a 3D QBT system or e.g., an LSM. The unusual temperature behavior of the dielectric constant indicates that the band structure of Pr227 evolves with temperature and the LSM ceases to be well defined above 100 K. Despite the presence of interactions that are almost two orders of magnitude more than the scale of the average kinetic energy, we find that the scattering rates below 80 K are Fermi-liquid like showing that at the lowest energy scales, a Fermi liquid state is stable when $E_F$ is finite. In this regard, the values of $E_F$ should be considered low as compared to the interaction strength $E_0$, but are still high as compared to the measured frequency range (and temperature range where the QBT effects are apparent). Our work raises issues as to the ultimate low temperature fate of such systems when accounting for doping and impurity effects. In future work, it would be interesting to further decrease the Fermi energy so that the interaction-dominated regime can be reached with both $\hbar\omega$ and $k_B T$ much less than $E_F$.

## Methods

**Film growth**. Two 200-nm Pr227 films (S1 and S2) were grown by pulsed laser deposition and annealing[28]. First amorphous precursor films were grown by pulsed laser deposition on yttria-stabilized zirconia YSZ(111) substrates. Then, epitaxial films were obtained by crystallizing the precursor layers by post-annealing in air at 1000 °C. The two films were found to be essentially identical with the only difference being slightly higher doping level in S2 that result in larger residual scattering, greater $E_F$, and a smaller contribution to the dielectric constant from the QBT. The optical experiment was performed by using a home-built time-domain terahertz spectrometer (TDTS) which can typically access the frequency range between 150 GHz and 2 THz[29]. Further details can be found in Methods. In the present experiment, the spectroscopic window was restricted from our typical range to frequencies below 1 THz by the intrinsically lossy substrates.

**Time-domain THz spectroscopy**. In TDTS, a femtosecond laser pulse is split along two paths and sequentially excites a pair of photoconductive Auston-switch antennae. A broadband THz range pulse is emitted by one antenna, transmitted through the sample under test, and measured at the other antenna. By varying the length difference of the two paths, the entire electric field of the transmitted pulse is mapped out as a function of time. The time-domain trace is then Fourier transformed into the frequency domain. Taking the ratio of the transmission through a sample to that of a reference resolves the full complex transmission coefficient. In the current case of thin films deposited on top of an insulating substrate, the transmission can be inverted to obtain the complex conductivity by using the appropriate expression in the thin film approximation[30].

**Drude-model fits**. To find the scattering rate and spectral weight of the lowest frequency features as well as the background dielectric constant, the optical conductivity data were fit to modified Drude/Drude–Lorentz model. In addition to the Drude term, we added a large dielectric term $\left(\frac{\tilde{\varepsilon}}{\epsilon_0}\right)$ that accounts for the background dielectric constant. As discussed in the main text, the frequency range of our measurements was greatly restricted from our usual one down to 0.3–1 THz due to YSZ substrate losses. In this region, only the Drude intraband transition is clearly resolved in the real part of the conductivity. Phonons are found at frequencies above 3 THz and give a negligible contribution to the spectra here. All data were fit to the below expression

$$\sigma(\omega) = \epsilon_0\left[-\frac{\omega_{\mathrm{p}}^2}{i\omega - 1/\tau} - i\left(\frac{\tilde{\varepsilon}}{\epsilon_0} - 1\right)\omega\right]. \quad (2)$$

Here $1/2\pi\tau$ is scattering rate of the Drude term, $\omega_{\mathrm{p}}$ is its plasma frequency, and $\epsilon_0$ is the vacuum permittivity. The background polarizability $\tilde{\varepsilon}$ originates from absorptions above the measured spectral range including phonons, local absorptions, and—in principle—the QBT.

We estimate the Fermi energy of our samples via the fitted Drude plasma frequency. The plasma frequency is given by the expression $\omega_{\mathrm{p}} = \sqrt{ne^2/m^*\epsilon_0}$ (SI unit). Up to constant factors, its square gives essentially the spectral weight (e.g., the area) of the Drude peak. The plasma frequency of sample S1 was found to be $\omega_{\mathrm{p}}/2\pi = 25.4$ THz and of sample S2 was $\omega_{\mathrm{p}}/2\pi = 33.8$ THz. Using the $m^* = 6.3m_0$ (where $m_0$ is the mass of a free electron) that has been reported in a recent photoemission study[20], we find that at the lowest temperature, charge densities for samples 1 and 2 are $n_1 = 5.1 \times 10^{19}$ cm$^{-3}$ and $n_2 = 8.7 \times 10^{19}$ cm$^{-3}$, respectively. Pr227 is a QBT system and we can assume that at small momenta, it has an isotropic Fermi surface. In three dimensions, Luttinger's theorem tells us that charge density and the Fermi wave vector ($k_F$) are related by the expression $n = k_F^3/3\pi^2$. Working from the above determined density, we find a $k_F$ of 0.11 Å$^{-1}$ for sample 1 and 0.14 Å$^{-1}$ for sample 2. The Fermi energy of the QBT system is $E_F = \hbar^2 k_F^2/2m^*$, where $m^*$ is the same as used above. The Fermi energies of samples 1 and 2 are then estimated to be $7 \pm 1$ meV and $12 \pm 1$ meV, respectively.

**Data availability**. All relevant data are available on request from N.P.A.

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

## Acknowledgements

We would like to thank L. Balents, P. Goswami, I. Herbut, Y.B. Kim, N. Laurita, E.-G. Moon, C. Varma, and L. Wu for helpful discussions. Experiments at JHU were supported by the Army Research Office Grant W911NF-15-1-0560. Work at ISSP was supported in part by CREST, Japan Science and Technology Agency, by Grants-in-Aid for Scientific Research (16H02209 and 26105002) and Program for Advancing Strategic International Networks to Accelerate the Circulation of Talented Researchers (No. R2604) from the Japanese Society for the Promotion of Science (JSPS), and by Grants-in-Aids for Scientific Research on Innovative Areas (15H05882 and 15H05883) from the Ministry of Education, Culture, Sports, Science, and Technology of Japan. S.N. had additional funding from a QuantEmX grant from ICAM and the Gordon and Betty Moore Foundation through Grant GBMF5305.

## Author contributions

B.C. performed and analyzed the TDTS measurements. Films were grown by T.O. and M. L. D.C. did the resistivity experiments with B.C. The manuscript was written by B.C. and N.P.A. with inputs from all authors. The project was directed by S.N., M.L., and N.P.A.

## Additional information

**Competing interests:** The authors declare no competing financial interests.

