## [Peer Review File · Nature Communications]

Reviewers' comments:

Reviewer #1 (Remarks to the Author):

The manuscript entitled "Giant dielectric anomaly and low frequency transport in a 3D quadratic band touching system: the Luttinger semimetal Pr₂Ir₂O₇" by B. Cheng et al. studies the transport properties of the conducting pyrochlore iridate Pr₂Ir₂O₇.

Pr₂Ir₂O₇ has recently received significant attention as having been suggested to represent a strongly correlated analogue of the inverted-band-gap material HgTe, potentially allowing to experimentally test predictions made by Abrikosov and Beneslavskii over 40 years ago, as well as a number of more recent theoretical results.

Using terahertz spectroscopy, Cheng and coworkers find a strongly temperature dependent dielectric constant that reaches an abnormally large value of $\tilde{\epsilon}/\epsilon_0 \sim 180$ at the lowest temperatures, which they argue to be consistent with an interacting quadratic band touching point (QBT) near the Fermi level.

The manuscript is very well written, and the conclusions appear to be scientifically sound. The interaction effects in zero-gap semimetals are subject of many theoretical work, but at the same time appear to be hardly accessible in experiment, at least in 3D. I therefore expect the results of Cheng et al. to be of immediate interest to many researchers in the field.

However, I do have a few comments which the authors should address in their manuscript in order to warrant, from my point of view, publication in Nature Communications.

(1) Besides the divergence of the dielectric constant at low temperatures (the functional form of which appears to be inconsistent with the simplest RPA, as the authors explain), the paper points out only few measured consequences of the strong-interaction physics characteristic of the QBT system. For instance, the temperature dependence of the plasma frequency ω_p is compared only to the prediction of the noninteracting QBT model, while the measurements are clearly done in a regime in which interactions are strong. In this regime, the dynamical critical exponent z should be smaller than 2, and presumably $\omega_p \sim T^x$ with an exponent $x = d/(2z) > 3/4$. Also, Ref. 16 predicts a scaling form for the optical conductivity in the strongly-interacting regime, $\sigma(\omega, T) \sim T^{1/z} F(\omega/T)$, with a universal scaling function F . It would be very interesting to test this scenario, e.g., by plotting $\sigma/T^{1/z}$ versus ω/T with suitably adjusted z .

(2) The authors claim that the Mott gap expected in Pr₂Ir₂O₇ according to theory "may be estimated to be more than 10 meV." As far as I understand, however, the expected Mott gap should be around $\sim 10^{-2} E_0$ [see Phys. Rev. B 95, 075101 (2017)], which would be significantly smaller than 10 meV, and might therefore be inaccessible in the present measurements. This may be even more true considering the non-negligible doping in the present samples. I suggest to tone down the statement.

(3) In the Introduction, it is stated that HgTe and α -Sn "can be tuned from a topologically trivial to a topologically non-trivial phase under uniaxial strain or in a superlattice geometry." I do not think this statement is correct. Protected edge states are present in HgTe also without external strain, uniaxial strain only lifts the cubic symmetry and opens up the bulk band gap.

(4) In the Supplementary Information, the authors claim that "both samples are actually very close to the touching point." I am not sure whether I can agree with this statement. To which energy scale is E_F compared to? Can the authors estimate the size of the impurity

concentrations?

(5) In Eq. (2) in the Supplementary Information, the argument of the step function appears to have the wrong sign.

Reviewer #2 (Remarks to the Author):

The paper "Giant dielectric anomaly.." describes the detailed temperature dependent optical response of the PR227 thin films in the THz frequency range, revealing temperature dependent dielectric constant, reaching large values (>100) at low temperatures. This unusual optical response is discussed in terms of Luttinger semimetal in terms of a 3D quadratic band touching system for temperatures below 100K.

These results and the interpretation of the data, to the best of my knowledge, are original with sufficient references to relevant theory and related experiments. Given the current interest in materials such as Weyl semimetals and the potential for such 3D QBT Luttinger semimetals to display more pronounced electronic correlations, this work would be of current interest to the condensed matter physics community. However, the interest of these results to researchers beyond, especially given the complexity of the analysis, is minimal.

The data and its analysis in terms of the determination of the THz conductivity is valid and described adequately in the supplementary material. However, this analysis could benefit from inclusion of an error analysis. However, the extrapolation of the solid spectral analysis to the interpretation and conclusions requires a significant amount of discussion, estimation and speculation and is not clear cut. This paper would benefit immensely from a more quantitative discussion of the results on pages 5-6 and should be a prerequisite for publication in Nature Communication. The paper as written is more suitable for a specialized journal such as Physical Review B.

Reviewer #3 (Remarks to the Author):

The authors report results of a THz spectroscopy study on thin film Pr₂Ir₂O₇. The central finding of this work is the large low-energy dielectric constant of Pr₂Ir₂O₇, which, according to the authors, suggests the virtual fluctuations in a 3D quadratic band touching system or a Luttinger semimetal.

The introduction of the manuscript is well written and the data are carefully taken. However, the manuscript is not free of speculations when data are interpreted.

Here are a few specific comments or questions:

1. It is essential for the electrical resistivity data for S2 sample to be presented along with that for S1 in Fig.1b in order to facilitate comparison.
2. Related to the comment above, Fig.3c shows a significant difference in the dielectric constant between S1 and S2 below 100 K. Since the central point of this work is made largely based on this set of data, the authors should address the difference.
3. The authors state that the scattering rates below 80 K is very Fermi-liquid like; and yet, the resistivity data below 80 K does not support a Fermi-liquid behavior at all. This significant discrepancy should be elaborated in the discussion.

Reviewer #1 (Remarks to the Author):

The manuscript entitled “Giant dielectric anomaly and low frequency transport in a 3D quadratic band touching system: the Luttinger semimetal Pr₂Ir₂O₇” by B. Cheng et al. studies the transport properties of the conducting pyrochlore iridate Pr₂Ir₂O₇.

Pr₂Ir₂O₇ has recently received significant attention as having been suggested to represent a strongly correlated analogue of the inverted-band-gap material HgTe, potentially allowing to experimentally test predictions made by Abrikosov and Beneslavskii over 40 years ago, as well as a number of more recent theoretical results.

Using terahertz spectroscopy, Cheng and coworkers find a strongly temperature dependent dielectric constant that reaches an abnormally large value of $\tilde{\epsilon} \wedge \epsilon_0 \sim 180$ at the lowest temperatures, which they argue to be consistent with an interacting quadratic band touching point (QBT) near the Fermi level.

The manuscript is very well written, and the conclusions appear to be scientifically sound. The interaction effects in zero-gap semimetals are subject of many theoretical work, but at the same time appear to be hardly accessible in experiment, at least in 3D. I therefore expect the results of Cheng et al. to be of immediate interest to many researchers in the field.

We appreciate the positive comments of the reviewer.

However, I do have a few comments which the authors should address in their manuscript in order to warrant, from my point of view, publication in Nature Communications.

(1) Besides the divergence of the dielectric constant at low temperatures (the functional form of which appears to be inconsistent with the simplest RPA, as the authors explain), the paper points out only few measured consequences of the strong-interaction physics characteristic of the QBT system. For instance, the temperature dependence of the

plasma frequency ω_p is compared only to the prediction of the noninteracting QBT model, while the measurements are clearly done in a regime in which interactions are strong. In this regime, the dynamical critical exponent z should be smaller than 2, and presumably $\omega_p \sim T^x$ with an exponent $x = d/(2z) > 3/4$.

We thank the referee for pointing this out. Naively, the linearity of the plasma frequency with temperature suggests then that $z \sim 3/2$. This value is indeed smaller than 2 as expected. However, we should point out that our system has finite E_F and the temperature dependent part of the plasma frequency is of the same order as the constant part so it's not clear that a power law should even be observed over our temperature range. Moreover, the plasma frequency that we are measuring here comes from the Drude contribution, which is also a result of having a (small) finite E_F . The resulting behavior even for the strongly interacting case, will likely not be just the sum of a constant and the "correct" power law. So upon reflection, we believe that our statement in our draft should be amended to point this out and that -- in the absence of a more complete theory -- comparison to any power law should be taken with grain of salt. We have made changes in this regards and no longer make a comparison the the non-interacting case.

Also, Ref. 16 predicts a scaling form for the optical conductivity in the strongly-interacting regime, $\sigma(\omega, T) \sim T^{1/z} F(\omega/T)$, with a universal scaling function F . It would be very interesting to test this scenario, e.g., by plotting $\sigma/T^{1/z}$ versus ω/T with suitably adjusted z .

We indeed had tried to use the scaling formula provided in Ref. 16 to collapse optical conductivity. But we found it difficult to fit by using this functional form in our case. The issue seems to again to be the effects of finite Fermi energy and residual impurity scattering. This can be seen as follows. As an empirical observation we have found that the optical conductivity can be fitted by the Drude model. The real part of the optical conductivity from the Drude model is:

$$\sigma_1(\omega) = \frac{\omega_p^2}{4\pi} \frac{1}{\frac{1}{\tau} + \omega^2}$$

By fitting the real parts of the conductivity, we found that $\omega_p = aT+b$, and $1/\tau = cT^2+d$. If we input these relationships into the Drude model above, we find

$$\sigma_1(\omega) = \frac{(aT + b)^2 (cT^2 + d)}{4\pi} \frac{1}{(cT^2 + d)^2 + \omega^2}$$

One can clearly see that if b and d are not zero, the formula cannot be simplified into a scaling form of a function that depends on ω/T .

Of course, one may expect that the scales introduced by impurity scattering and finite E_F may be overcome at high enough frequency and/or temperature scales. The

problem is that these frequency scales are not accessible to us (due to the lossy substrate) and it appears that at higher temperature scales the quadratic band touching physics breaks down (as evidenced by the temperature dependence of our dielectric constant and the loss of the well-defined bands in the ARPES spectra above 100 K).

We believe we have evidence for non-trivial consequences of the quadratic band touching physics through the phenomenology of the dielectric constant and scattering rate, but we don't believe a detailed collapse of the conductivity with a scaling form is possible in the current experiment.

(2) The authors claim that the Mott gap expected in Pr2Ir2O7 according to theory “may be estimated to be more than 10 meV.” As far as I understand, however, the expected Mott gap should be around $\sim 10^{-2} E_0$ [see Phys. Rev. B 95, 075101 (2017)], which would be significantly smaller than 10 meV, and might therefore be inaccessible in the present measurements. This may be even more true considering the non-negligible doping in the present samples. I suggest to tone down the statement.

The referee is correct that our number was off, but we don't believe it is off by that much. We estimate that E_0 is of order 0.41 eV, so by the expression given in that PRB this Mott scale should be about 4 meV. This is ~ 1 THz which is within our measurement range, but is also smaller than the inferred E_F s. And this will be relevant as the referee points out. At any rate, this issue is not essential to our principle focus so we have toned down this discussion.

(3) In the Introduction, it is stated that HgTe and alpha-Sn “can be tuned from a topologically trivial to a topologically non-trivial phase under uniaxial strain or in a superlattice geometry.” I do not think this statement is correct. Protected edge states are present in HgTe also without external strain, uniaxial strain only lifts the cubic symmetry and opens up the bulk band gap.

The referee is correct. We miswrote, and have now corrected it to just say that strain can push HgTe into a gapped TI phase.

(4) In the Supplementary Information, the authors claim that “both samples are actually very close to the touching point.” I am not sure whether I can agree with this statement. To which energy scale is E_F compared to? Can the authors estimate the size of the impurity concentrations?

In supplementary information, we discuss how we estimate Fermi energy of samples S1 and S2 from the Drude spectral weight and the band parameters (from ARPES). We find values of 7.3 meV and 11.9 meV respectively. The referee's question about to what scale E_F can be compared to is a good one. One relevant scale is E_0 , which is the characteristic scale of the electron-hole interaction. This is the scale below which interactions can be considered strong and the interaction physics of the quadratic band touching may manifest. As discussed above E_0 is of order 0.4 eV and so this is at least one sense in which we can consider E_F small and “close” to the touching point.

However, the low temperature scales where the QBT physics manifests is smaller than EF. And our probing frequencies are smaller than EF. And so by these standards EF is indeed large. This is exactly the reason why the scaling form discussed above does not work. We have rewritten this section to say exactly what we mean and what the considerations are.

We don't know the impurity concentration per se, but the residual dopant level is $\sim 10^{19} \text{cm}^{-3}$ and so this sets some rough estimate on the deviations from non-stoichiometry.

(5) In Eq. (2) in the Supplementary Information, the argument of the step function appears to have the wrong sign.

We have corrected this.

Reviewer #2 (Remarks to the Author):

The paper "Giant dielectric anomaly.." describes the detailed temperature dependent optical response of the PR227 thin films in the THz frequency range, revealing temperature dependent dielectric constant, reaching large values (>100) at low temperatures. This unusual optical response is discussed in terms of Luttinger semimetal in terms of a 3D quadratic band touching system for temperatures below 100K. These results and the interpretation of the data, to the best of my knowledge, are original with sufficient references to relevant theory and related experiments.

We appreciate the positive comments from the referee.

Given the current interest in materials such as Weyl semimetals and the potential for such 3D QBT Luttinger semimetals to display more pronounced electronic correlations, this work would be of current interest to the condensed matter physics community. However, the interest of these results to researchers beyond, especially given the complexity of the analysis, is minimal.

We appreciate that the referee believes that it will be of interest to the condensed matter physics community and we agree that our work should be seen as important in the context of interest in exotic semimetals and here in the enhanced correlations. But we disagree with the referee on two other points. First, we do not think that the analysis should be regarded so complex. Our optical conductivity observations are straightforward and we have analyzed the spectra in terms of conventional Drude transport and interpreted the physics in terms of a minimal model of the quadratic band touching. Persons familiar with optical spectra and Drude transport should find the manuscript straightforward.

Moreover, although we are not sure that it is a necessary requirement for Nature Communications, the physics in our paper *should* have resonance in the larger field of people working in strongly interacting systems and not just solid-state physics.

Similarly, the physics of Weyl and Dirac systems and graphene has sparked interest in field theorists of a particle physics persuasion. We expect that as QBTs are an example of a tractable strongly interacting system there should be interest in the present system in this regard. Indeed we noticed there were talks on interactions in quadratic band touching systems at the field theory community's "Functional Renormalization" and "New Developments in Conformal Field Theory Above Two Dimensions" meetings this year among others. However, one may speculate that the reason why there has not been overwhelming interest so far is that there have been no experiments. Our paper changes this.

The data and its analysis in terms of the determination of the THz conductivity is valid and described adequately in the supplementary material. However, this analysis could benefit from inclusion of an error analysis.

Some of the values we have determined already did have errors assigned, but we agree with the referee that all important values should have errors. They have been added in the revised version of the paper.

However, the extrapolation of the solid spectral analysis to the interpretation and conclusions requires a significant amount of discussion, estimation and speculation and is not clear cut. This paper would benefit immensely from a more quantitative discussion of the results on pages 5-6 and should be a prerequisite for publication in Nature Communication. The paper as written is more suitable for a specialized journal such as Physical Review B.

Unfortunately, we do not understand what the referee is requesting here. He or she seems to think that there is already too much discussion, e.g. that the conclusions "requires a significant amount of discussion, estimation and speculation", but then is requesting even more discussion of page 5 and 6. And the referee says they want more quantitative discussion, but we don't know what this would even be (aside from the error analysis we have now included). We have extracted all quantities we can and discussed them thoroughly. Page 5 has the quantitative discussion of corrections to the exponents and a discussion of the explicit values of the prefactor to the T^2 term and page 6 is the concluding remarks, acknowledgements and references. We should mention, that we wrote the manuscript in a style with "a significant amount of discussion" not because we thought it was necessary, but because we thought it would be interesting to readers. We still think so. In this regard, both Referees 1 and 3 said that the paper was "very well written".

We do not think that the paper is only suitable for PRB. The manuscript was written to make the manuscript broadly accessible, the subject matter is very topical due to current interest in zero gap semimetals (100s of papers), and goes beyond this subject to discuss the role of interactions. Moreover, (as Referee 1 points out) our manuscript accesses physics that has been impossible to get out in other fashion. We hope that the referee will now agree.

Reviewer #3 (Remarks to the Author):

The authors report results of a THz spectroscopy study on thin film Pr₂Ir₂O₇. The central finding of this work is the large low-energy dielectric constant of Pr₂Ir₂O₇, which, according to the authors, suggests the virtual fluctuations in a 3D quadratic band touching system or a Luttinger semimetal.

The introduction of the manuscript is well written and the data are carefully taken. However, the manuscript is not free of speculations when data are interpreted.

We appreciate the positive remarks of the referee. We hope to assuage the referee's concerns below as we believe that the points that he or she thinks is speculation may have arisen in a misunderstanding.

Here are a few specific comments or questions:

1. It is essential for the electrical resistivity data for S2 sample to be presented along with that for S1 in Fig.1b in order to facilitate comparison.

We agree with the referee and so we have added the dc resistivity of S2 to Fig.1b.

2. Related to the comment above, Fig.3c shows a significant difference in the dielectric constant between S1 and S2 below 100 K. Since the central point of this work is made largely based on this set of data, the authors should address the difference.

As we discuss in the manuscript, this is due to small differences in the doping level between the films. One of the significant aspects of our work is the self-consistency of the analysis e.g. the sample with the higher Fermi energy shows a lower dielectric constant. Indeed, we find even quantitative consistency between these values.

In the main text, we explain the reason for the difference of the dielectric constants of S1 and S2. In the undoped Luttinger semimetal, the dielectric constant is expected to diverge as $\sim \omega^{-0.5}$. However, our Pr₂Ir₂O₇ thin films are actually slightly hole doped and their Fermi level is not zero. Therefore the divergence of dielectric constants will be removed by this finite Fermi level because the interband transition excitations below $2E_F$ will be blocked. As we explained in the manuscript, the low-energy dielectric constant will be proportional to $(2E_F)^{-0.5}$. We have provided the detailed formulas in the main text. From the discussion above and in the main text, one can see the dielectric constants we measured in THz region are strongly dependent on the Fermi energies. We have used our optical data and some dc transport data to estimate the Fermi energies. The Fermi energies of S1 and S2 we find from the Drude spectral weight are found to be 7.3 meV and 11.9 meV respectively, which (as we discuss) allows us to independently predict the dielectric constant to high accuracy. There is a beautiful self-consistency between different independent parts of the data.

3. The authors state that the scattering rates below 80 K is very Fermi-liquid like; and

yet, the resistivity data below 80 K does not support a Fermi-liquid behavior at all. This significant discrepancy should be elaborated in the discussion.

As we discuss in the manuscript one of the many issues we shed light on is the long discussed anomalous temperature dependence of the resistivity of Pr227. We show that its unusual non-monotonic behavior derives from the competing temperature dependencies of decreasing carrier density and decreasing scattering rate.

An essential property of a Fermi liquid is that its scattering rate goes as $\sim T^2$. In a conventional metal, the free carrier density does not depend on temperature, which results in a T^2 dc resistivity. However, in the present case, these semimetals have a very strongly temperature dependent charge density. Previous dc resistivity studies of Pr227 found its dc resistivity has an upturn around 50 K, and concluded this behavior comes from the Kondo effect. However, the dc resistivity measures the joint effects of charge density and scattering rates and cannot isolate their separate dependencies. In our work, by fitting the optical conductivity, the charge density and scattering rate can be isolated and we show in Pr227, that the charge density goes as $\sim T$ and scattering rates is $\sim T^2$. Although the dc resistivity does not show T^2 behavior, the more fundamental quantity of the scattering rate show $\sim T^2$ down to very low temperature, which is an indication of a Fermi liquid. In this regard the THz measurements are essential in determining the Fermi liquid nature of the material. We should reiterate that this Fermi liquid nature is likely only arising due to finite EF. For EF=0 either a non-Fermi liquid or an ordered state is likely to result.

REVIEWERS' COMMENTS:

Reviewer #1 (Remarks to the Author):

From my point of view, the issues raised in the previous round of review have been satisfactorily addressed, except that I had not been able to locate any rewriting of Section S1, saying "exactly what we mean and what the considerations are" concerning the size of E_F , as the authors claim in their reply. Have they added any comment in replacement to their previous statement that "the Fermi levels of both samples are actually very close to the touching point"? If not, this should be rectified.

In summary, I still adhere to my original judgment that the manuscript is well written in a clear and understandable way, and the conclusions drawn appear to be scientifically sound. From my perspective, the only major drawback of this work is the still not-so-small residual dopant level present in the samples, preventing a controlled access to the even more exotic very-low-energy physics expected in this material (scale invariance, nontrivial exponents, Mott transition, ...). However, given the fact that the manuscript represent the first work clearly demonstrating the very presence of strong interaction effects in a three-dimensional quadratic band touching point system, I find the results obtained to be of sufficient broad interest to warrant publication in Nature Communications.

Reviewer #2 (Remarks to the Author):

I agree that the authors have addressed my concerns and that the manuscript is suitable for publication in Nature Communications.

Reviewer #3 (Remarks to the Author):

The revised manuscript is improved. To some extent, the authors' response addresses the concerns raised by this referee.

The topic of this work is among the most current and interesting ones. However, this referee remains unconvinced that this particular work is suitable to the general audience and, equally important, this work clearly illustrates any novelty that merits publication in this journal.

Reply to Referees

Reviewer #1 wrote

"From my point of view, the issues raised in the previous round of review have been satisfactorily addressed, except that I had not been able to locate any rewriting of Section S1, saying "exactly what we mean and what the considerations are" concerning the size of E_F , as the authors claim in their reply. Have they added any comment in replacement to their previous statement that "the Fermi levels of both samples are actually very close to the touching point"? If not, this should be rectified."

The referee is correct. Somehow this text got edited out of the previous version. We apologize for this. We have put a few sentences in the conclusion that reads

"In this regard the values of E_F should be considered low as compared to the interaction strength E_0 but are still high as compared to the measured frequency range (and temperature range where the QBT effects are apparent). Our work raises issues as to the ultimate low temperature fate of such systems when accounting for doping and impurity effects. In future work it would be interesting to further decrease the Fermi energy so that the interaction dominated regime can be reached with both $\hbar\omega$ and $k_B T$ much less than E_F ."

We believe that this makes the point that the referee wanted and apologize for the oversight.